# First Case Report of Maternal Mosaic Tetrasomy 9p Incidentally Detected on Non-Invasive Prenatal Testing

**DOI:** 10.3390/genes12030370

**Published:** 2021-03-05

**Authors:** Wendy Shu, Shirley S. W. Cheng, Shuwen Xue, Lin Wai Chan, Sung Inda Soong, Anita Sik Yau Kan, Sunny Wai Hung Cheung, Kwong Wai Choy

**Affiliations:** 1Department of Obstetrics and Gynaecology, Pamela Youde Nethersole Eastern Hospital, Chai Wan, Hong Kong, China; clw042@ha.org.hk; 2Clinical Genetic Service, Hong Hong Children Hospital, Ngau Tau Kok, Hong Kong, China; Shirley.s.cheng@gmail.com; 3Department of Obstetrics and Gynaecology, Chinese University of Hong Kong, Hong Kong, China; xuesw@link.cuhk.edu.hk; 4Department of Clinical Oncology, Pamela Youde Nethersole Eastern Hospital, Chai Wan, Hong Kong, China; Soongs@ha.org.hk; 5Prenatal Diagnostic Laboratory, Tsan Yuk Hospital, Sai Ying Pun, Hong Kong, China; kansya@hku.hk; 6NIPT Department, NGS Lab, Xcelom Limited, Hong Kong, China; sunny.cheung@xcelcom.com

**Keywords:** mosaicism, tetrasomy 9p, non-invasive prenatal test, normal phenotype

## Abstract

Tetrasomy 9p (ORPHA:3390) is a rare syndrome, hallmarked by growth retardation; psychomotor delay; mild to moderate intellectual disability; and a spectrum of skeletal, cardiac, renal and urogenital defects. Here we present a Chinese female with good past health who conceived her pregnancy naturally. Non-invasive prenatal testing (NIPT) showed multiple chromosomal aberrations were consistently detected in two sampling times, which included elevation in DNA from chromosome 9p. Amniocentesis was performed and sent for chromosomal microarray, which was normal. Maternal karyotype revealed that mos 47,XX,+dic(9;9)(q21.1;q21.1)(24)/46,XX(9) presents mosaic tetrasomy for the short arm of chromosome 9p and is related to the NIPT results showing elevation in DNA from chromosome 9p. The pregnancy was uneventful, and the patient was delivered at term. Maternal samples were obtained at two different time points after delivery showed the same multiple chromosomal aberrations detected during pregnancy. This is a first report on an unusual case of mosaic isodicentric tetrasomy 9p in a healthy adult with normal intellect. With widespread adoption of NIPT for screening fetal aneuploidy and genome-wide copy number changes, a rise in incidental detection of maternal rare genetic syndrome will bring challenges in our current approach to genetic counselling and prenatal diagnosis.

## 1. Introduction

Tetrasomy 9p was first reported by Ghymers et al. in 1973 [1]. It is a rare chromosomal syndrome with typical features such as severe psychomotor retardation, skeletal and renal abnormalities, congenital heart disease, and facial dysmorphism [2]. About 30% of living cases exhibit mosaicism [3]. Of all the reported cases with tetrasomy 9p mosaicism, approximately 50% of them show a characteristic facial appearance, growth impairment, and developmental retardation [4]. The milder end of the disease spectrum has been shown with tissue-specific mosaicism of tetrasomy 9p, postulating that the severity of the phenotype may be related to the degree of mosaicism and the extent of tissues involvement [5]. Rarely, patients with mosaic tetrasomy 9p that present with no apparent clinical features are described in the literature [6,7,8].

Here we report of a pregnant woman with tetrasomy 9p mosaicism discovered by non-invasive prenatal testing (NIPT), also known as cell free fetal DNA (cffDNA) testing. The results of the molecular, clinical, and cytogenetic findings are compared to previous case reports published.

### Case Report

A 33-year-old nulliparous women was referred to prenatal diagnostic counselling for two non-reportable non-invasive prenatal tests (NIPT) despite sufficient total fetal fraction of 7 and 6.4%, respectively. Both samples showed multiple chromosomal aberrations, which included elevation in DNA from chromosome 9p, and reductions in DNA from chromosomes 1, 2, 3, 4, 5, 6, 7, 8, 11, 12, 13, 14, 18, 20, and 21. Amniocentesis was performed at 17 weeks gestation, and chromosomal micro array revealed a normal male fetus. Maternal karyotyping reveals mosaic tetrasomy for the short arm of chromosome 9 and is related to the NIPT result, showing elevation in DNA from chromosome 9p. After genetic counselling, subtelomeric multiplex ligation-dependent probe amplification (MLPA) was performed on buccal swab and on uncultured blood. The buccal swab result was normal, but the blood test confirms mosaic 9p duplication.

The patient had a high level of education and is a dentist by profession. She has no family history of genetic abnormalities or developmental delay. Her twin brother and elder sister are healthy. This pregnancy was conceived naturally. Interestingly, she was worked up for infertility a year prior to conception and was found to have a low anti-Mullerian hormone level (5.64 pmol/L), which predicts low ovarian reserve. On physical exam, Blanschko’s line was evident over the back, the abdomen, the forearms, and lower limbs (Figure 1a,b). Her ears appeared low set and there were also multiple locks of white hair on the front and sides of her head. She delivered a health boy weighing 3035 g at term by Caesarean section. The placenta was submerged in formalin before it could be saved for genetic investigation. Repeated NIPT were performed at 6 days and at 1 month post-delivery, which yielded the same NIPT results with multiple aberrations over many chromosomes and includes elevation in DNA from chromosome 9p. The elevated signal in chromosome 9p is consistent with mosaic tetrasomy 9p reported in maternal karyotype. However, whether the reduction in DNA from the other chromosomes are contributed to by the mother or other source of interference will need further investigations.

## 2. Materials and Methods

Peripheral blood for NIPT was done over four time points. It was taken during pregnancy at 11 weeks and 15 weeks gestational age (ga), and it was repeated post-delivery day 6 and post-delivery day 30. Ten milliliters of maternal peripheral blood were collected into a Streck BCT tube and proceeded for genome-wide cffDNA by using the protocols as previously reported [9]. All procedures and molecular tests, including cell-free DNA isolation, library construction, sequencing, and bioinformatics analyses, were performed at the Prenatal Genetic Diagnosis Laboratory of the Department of Obstetrics and Gynaecology, The Chinese University of Hong Kong. Following library construction and amplification, the samples were sequenced on the Nextseq500 (Illumina, San Diego, CA, USA) with a minimum of 20 million read pairs per sample. The scope of detection and reporting of the genome-wide cffDNA screening included risk assessment for trisomy 21, trisomy 18, trisomy 13, sex chromosomal abnormalities, and genome-wide chromosomal aberrations at a resolution of 3Mb or above. Combined count-based and size-based analyses were adopted for the detection of chromosomal aberrations. Pretest counselling was provided by obstetrician and written consent was obtained from the patient.

Following two non-reportable NIPT results at 11 and 15 weeks ga, conventional cytogenetics was performed by Giemsa banded (G-banded) karyotyping on cultured lymphocytes from maternal blood sample at Prenatal Diagnostic Laboratory, Tsan Yuk Hospital. Peripheral blood was collected in heparinized tubes at 17th weeks. Culturing and harvesting of peripheral blood lymphocytes were performed according to standard protocol.

The patient was seen by a clinical geneticist at 21 weeks ga. Consent obtained and buccal swabs and blood samples were collected for subtel MLPA testing at Clinical Genetic Service (Department of health). DNA was extracted from buccal swab and blood sample for subtel MLPA testing using the SALSA Probemix P036 Subtelomeres Mix 1 and P070 Subtelomeres Mix 2B kits (MRC-Holland, Amsterdam, The Netherlands), which were designed to detect deletion or duplication of the subtelomeric regions of all chromosomes.

The subject gave her informed consent for inclusion before she participated in the study. The study was conducted in accordance with the Declaration of Helsinki, and the protocol was approved by the Joint Chinese University of Hong Kong-New Territories East Cluster Clinical Research Ethics Committee (Project identification code 2017.168).

## 3. Results

### 3.1. NIPT Identified Multiple Copy Number Changes

Genome-wide cffDNA screening was initially performed at 11 weeks ga (fetal fraction: 7.0%) and showed multiple reductions in DNA from chromosomes 1, 2, 3, 4, 5, 6, 7, 8, 11, 12, 13, 14, 18, 20, and 21, as well as elevated amounts of DNA from the p-arm of chromosome 9 with a mean count-based z-score of 149.8 and aberrant region (9p24.3-9p13.1) z-score of 360.84, respectively. The finding on chromosome 9 suggested a gain of approximately 38.5 Mb in size encompassing chromosome bands 9p24.3-9p13.1 (GRCh37:Chr9:1-38,500,000). Based on the high z-score, the increased chromosome 9p signal could be contributed to by the mother.

Genome-wide cffDNA was repeated at 15 weeks ga (fetal fraction: 6.4%), post-delivery day 6 days and post-delivery day 30. The results of multiple chromosomal aberrations detected were highly consistent with the original NIPT results at 11 weeks ga, resulting in an identical increase on chromosome 9 p arm with consistently high count-based z-score, and aberrant region (9p24.3-9p13.1) z-score range over 100. There were identical profiles observed across the entire genome except chromosome 9 (Figure 2).

### 3.2. Karyotyping

Conventional cytogenetic analysis of cultured lymphocytes from maternal blood, collected at 17 weeks ga, showed mosaic female karyotype with the presence of two cell lines. Cell line 1 (24/33; 73% cells) has two normal chromosome 9 and dicentric chromosome with breakage and union in the long arm of chromosome 9 at band 9q21.1 (Figure 3). C-banding shows two dark bands in the middle of the dicentric chromosome, indicating the presence of two heterochromatic regions. This presents partial tetrasomy of chromosome 9 from 9pter to 9q21.1. Cell line 2 (9/33; 27% cells) has a normal female karyotype. The proportion of the cell lines may be different in other tissues.

### 3.3. Subtel MLPA

Subtel MLPA was performed on blood and buccal swab at 21 weeks gestational age. Heterozygous duplication of 9p subtelomeric region was found on blood samples but not the buccal swab. The result indicates somatic mosaicism of 9p duplication in this patient.

## 4. Discussion

Tetrasomy 9p has been described in two forms: isodicentric chromosome 9p (i(9p)), where the two short arms are linked by a single centromeric region, and pseudodicentric 9p (idic(9p)), with the breakage and reunion in the long arm linking one active and one inactive centromere [10]. For the latter, the term dicentric 9p(dic(9p)) is sometimes used. In living patients, i(9p) and idic(9p) usually present in mosaic states with variable phenotypes. In the worst case scenario with mosaic state, one could have neurodevelopmental delay with metabolic disorder [11]. Individuals having no or mild features are rarely encountered [6,7,8,12,13,14], and here we present the eighth literature report on a patient with mosaic tetrasomy 9p exhibiting mild phenotype and normal intelligence. To our knowledge, she is also the first maternal case with the unique dicentric variety that was discovered by cell free fetal DNA prenatal screening test.

Table 1 presents a summary on the indications for the genetic tests, findings, and outcomes of the eight cases. Fertility-related issues were a common theme (*n* = 5), one being premature ovarian failure. In the first case [6], investigation was prompted by skin lesions peculiar to mosaicism. Two cases [8] that were discovered incidentally had a completely normal phenotype. The majority of small supernumerary marker chromosome carriers (approximately 2 million individuals) will never know about their condition and may learn of it by chance [15]. The reasons for the phenotypical variability presented by mosaic tetrasomy 9p are largely unknown. The level of mosaicism may have a role. Small supernumerary marker cases that were thought to carry poor outcome but instead presented with mild phenotype have been reported with isochromosome 18p [16] and 12p [17]. For the eight cases mentioned in Table 1, they all have a relatively normal phenotype with level of mosaicism ranging from 6–100%. In our case, mosaicism was observed in 73% of the peripheral blood cells and the buccal swab was normal. It was also the only case with dicentric 9p. We hypothesize that the extent of tissue involvement, the point of breakage, and reattachment may affect the phenotypical variability.

With our case, low anti-Mullerian hormonal level was detected in routine work up for infertility. She was warned about the possibility of premature ovarian failure, and the impact of future fertility Premature ovarian failure is defined by the cessation of ovarian function before 40 years old and is related to low anti-Mullerian hormone level [18]. Although, X chromosome abnormality is the commonest cytogenetic cause of premature ovarian failure [12], other genetic causes are yet to be determined. As far as male fertility is concerned, a recent case report showed that i(9p) was not found in spermatozoa [13].

Blanschko’s line, which are cutaneous manifestations of mosaicism [19], was observed in our patient. The skin provides an opportunity to visualize and study patterns of mosaicism in a way that is not possible for internal organs. However, urine test offers unique access to examine the bladder cell lines. The level of mosaicism is usually stronger in blood samples compared to the skin and buccal swabs [20].

NIPT has been widely used to screen for common aneuploidies since 2011 with sensitivity above 99% for trisomy 21, with test failure at up to 12.7% [21]. Maternal mosaicism is a well-known contribution for discordant results leading to test failure, as demonstrated in our case report. Twenty-two antenatal cases of tetrasomy 9p have been reported to date, which were all discovered by invasive cytogenetic testing due to severe congenital abnormalities [22]. Around 68% (15/22) of those cases were mosaic state and one survived past the neonatal period. A majority were terminated or succumbed to intrauterine death. Therefore, fetal tetrasomy 9p, whether full or mosaic, is associated with poor prognosis.

Confined placental mosaicism is a well-known cause of aberrant NIPT for aneuploidy [23] that can be ruled out with multiple biopsies of the placenta after delivery. Placenta biopsies could not be saved. Reassuringly enough, fetal DNA is rapidly cleared from maternal circulation and is undetectable by 2 h after delivery [24]. Fetal DNA found in maternal blood is derived from the placenta. By repeating NIPT post-delivery at 6 days and 30 days, we could confidently assume that any influence from the placenta was ruled out. Indeed, the chromosomal aberrations persisted, confirming once again it was maternal origin.

Another cause of discordant NIPT results is maternal malignancy, with a cancer risk of 1 in 1000 pregnant women [25]. Tumour cells shed DNA into the maternal circulation and are picked up by NIPT. This discovery has opened a new field for molecular diagnosis of cancer [26]. For our patient, persistent aberrant findings of reduction in DNA from chromosomes 1, 2, 3, 4, 5, 6, 7, 8, 11, 12, 13, 14, 18, 20, and 21 lead to concerns of malignancy. However, a consensus has not been reached on how to screen for malignancy in such cases. The approach suggested by Carlson et al. [27] was adopted for our patient and showed no evidence of malignancy.

Our patient delivered a healthy male baby of normal birth weight by Caesarean section at term. As the amniocentesis was normal, further genetic testing for the child was not done. Implication on future pregnancy was discussed with our patient. Should she wish to have Down Syndrome screening for her next pregnancy, a traditional method consisting of nuchal translucency measurement with biochemical marker would be more suitable. NIPT in her next pregnancy will inevitably be non-reportable again.

## 5. Conclusions

NIPT is becoming widely accessible and its scope of use is expanding in the commercial world. Inevitably, this will open a Pandora’s box of issues that needs further consideration. Our patient experienced an unexpected revelation twice during her pregnancy; learning that she is a carrier of a rare genetic imbalance, and potentially harbouring a malignancy that is yet unproven. As this technology expands to include early detection of malignancy as well, the implications are no longer confined to women of child-bearing age. Incidental detection of rare genetic conditions is likely to increase in the era of cell-free DNA, bringing with it a wealth of new knowledge and conundrums. The onus is upon the clinicians to provide proper pre- and post-test counselling, which includes incidental findings.

## Figures and Tables

**Figure 1 genes-12-00370-f001:**
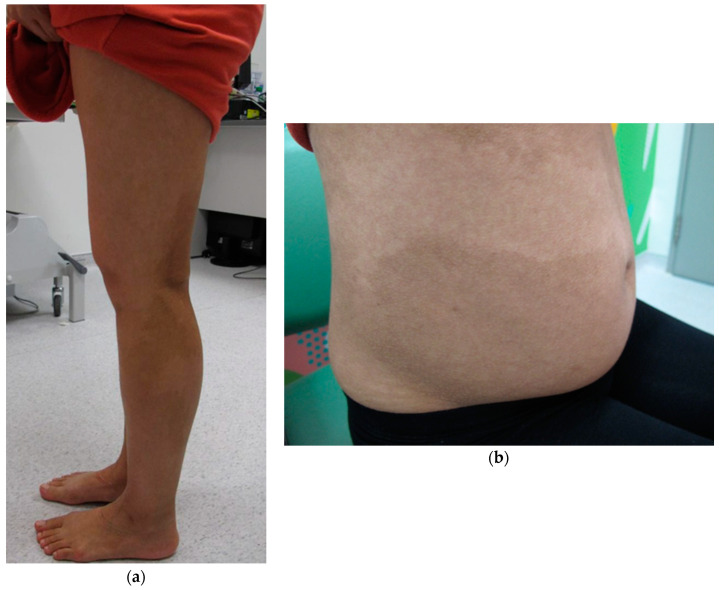
Blanschko’s line is evident on (**a**) the lower limbs and (**b**) over the abdomen; photograph taken when the patient was pregnant at 25 weeks of gestation.

**Figure 2 genes-12-00370-f002:**
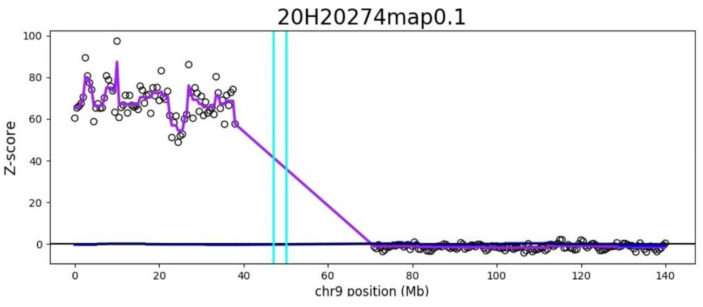
Detection of increased counts of DNA originating from chromosome 9p by genome-wide cfDNA screening post-delivery day 6. The genomic position is shown on the x-axis and the count-based z-score is shown on the y-axis. Each open circle on the plot represents a 1-Mb window. The light blue–green vertical lines mark the boundaries of the centromere. Genome-wide cfDNA screening results: increased DNA from 9p24.3-p13.1:1-38,500,000, count-based z-score of 70.

**Figure 3 genes-12-00370-f003:**
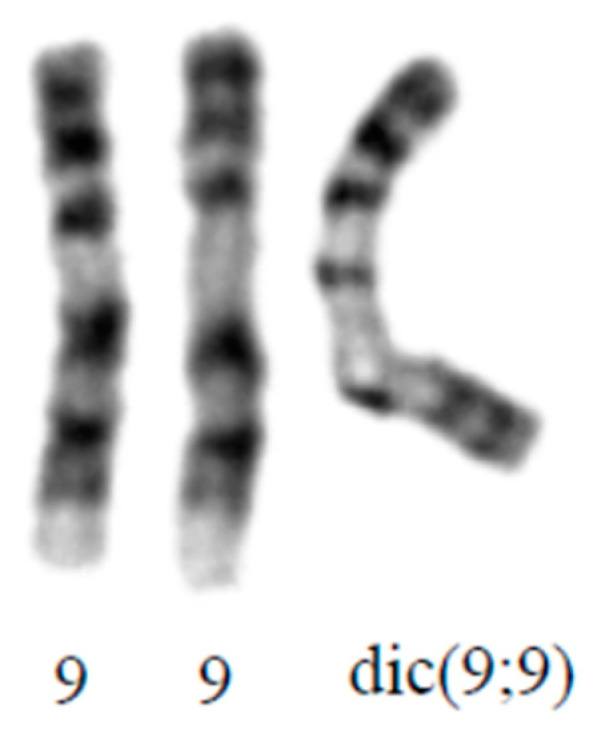
Partial karyogram of cultured lymphocytes from peripheral blood of patients showing dic(9;9)(q21.1;q21.1).

**Table 1 genes-12-00370-t001:** Cytogenetic results and clinical data in mosaic tetrasomy 9.

		Sait and Wetzler (2003)	McAuliffe et al., (2005)	Ogino et al., (2007)	Baronchelli et al., (2011)	Papoulidis et al., (2012)		Bellil et al., (2019)	Present Case (2021)
						Case 1	Case 2		
Age		41	37	10	Adult	20	28	41	33
Sex		Male	Male	Male	Female	Female	Female	Male	Female
Indication for karyotyping		Skin lesions	Oligozoospermia	Klinefelter-like phenotype	Premature ovarian failure	Familial inv. (7)	IVF for male azoospermia	Oligozoospermia IVF failure	Abnormal NIPT
Phenotype		Healthy Hypereosinophilia in BM, peripheral blood, skin lesion	Healthy	Healthy Concealed penis	Healthy	Healthy	Healthy	Healthy	Healthy Blanschko’s line, Low set ears, Hypopigmented hair, Low ovarian reserve
Supernumerary variety		i(9p)	i(9p)	i(9p)	i(9p)	i(9p)	i(9p)	i(9p)	dic(9p)
Mosaicism level	Blood	43%	20%	6%	72%	100%	80%	50–60%	73%
	Skin	Unknown	0%	Unknown	Unknown	Unknown	Unknown	Unknown	Unknown
	Buccal mucosa	Unknown	Unknown	5%	Unknown	65%	Unknown	6%	0%
	Other tissue	Bone marrow 86%	Unknown	Unknown	Unknown	Unknown	Unknown	Unknown	Urine 50% Sperm 0%
Offspring	Number	Unknown	5 miscarriages 2 neonatal deaths 2 children	No	No	1 child	Unknown	1 miscarriage	1 child
	Karyotypes	-	One trisomy 15 and 4 normal	-	-	Unknown	-	-	Normal
Origin		Unknown	Unknown	Unknown	Unknown	Unknown	De novo	Unknown	Unknown

NIPT = Non-invasive prenatal test; i(9p) = isodicentric chromosome 9p; dic(9p) = dicentric chromosome 9; BM = Bone Marrow.

## Data Availability

Data sharing not applicable. No new data were created or analyzed in this study. Data sharing is not applicable to this article.

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
