# Peer review of "First Case Report of Maternal Mosaic Tetrasomy 9p Incidentally Detected on Non-Invasive Prenatal Testing"

_genes, 2021, doi:10.3390/genes12030370_

Round 1

Reviewer 1 Report

The Authors report a case of rare maternal mosaic tetrasomy 9p detected during non-invasive prenatal testing. The report represents an interesting medical genetic work, worth of diffusion in the field of human genetics/genetic testing.

Being a case report, it has a purely descriptive nature and basic methods. However, the rarity of the genetic condition and the rigorous way it was studied, reported and discussed make it a piece of work deserving full consideration for publication.

I do not have major concerns. However, I would suggest a language revision, since some sections present various typos.

In addition I believe that the legend of figure 3: A) Karyogram showing... should be corrected with: Karyotype showing... 

Author Response

Reviewer 1

Point 1: In addition I believe that the legend of figure 3: A) Karyogram showing... should be corrected with: Karyotype showing... 

Response 1: We revised the legend of figure 3 as below:

"Partial karyogram of cultured lymphocytes from peripheral blood of patient showing +dic (9;9)(q21.1;q21.1)"

We use karyogram for digital or pictorial representation of the chromosome, whereas karyotype is the nomenclature.

Point 2: I do not have major concerns. However, I would suggest a language revision, since some sections present various typos.

Response 2: noted 

Reviewer 2 Report

This is an interesting case report on tetrasomy 9p. 

The abstract already summarises and explains briefly the case.

However, there is some missing link between material method, result, and discussion, also the conclusion.

Line 106 and 114-116 should also be added in material and methods.

There is some inconsistency between material methods and results:

Material method line 95 (peripheral blood collection at 17 weeks pregnancy)

Maternal blood sample and buccal swab at 21 weeks pregnancy.

While the results showed:

screening at 11+3 weeks then additional at 15+4 weeks of gestation and 6 days and 1 month after delivery.

How about the results from maternal blood sample and buccal swab?

The discussion should discuss the results compare to the available literature.

The conclusion should be expanded and more comprehensive as it was already reflected in the abstract.

Since the baby was already born, further perinatal information should be included.

Author Response

Response to reviewer 2

Point 1: Line 106 and 114-116 should also be added in material and methods.

Response 1: Line 106 was already in the material and method section. Line 114-116 describes the results of the genome wide cffDNA test and should be in the results section.

Point 2: There is some inconsistency between material methods and results: Material method line 95 (peripheral blood collection at 17 weeks pregnancy)

Maternal blood sample and buccal swab at 21 weeks pregnancy.

While the results showed:

screening at 11+3 weeks then additional at 15+4 weeks of gestation and 6 days and 1 month after delivery.

How about the results from maternal blood sample and buccal swab?

response 2: I have rounded off the gestational age to weeks and omitted the days for the benefit of simplicity. Be assured that the results are not changed by this omission. NIPT (also called cffDNA) was done at 11 weeks gestational age (ga) was non-reportable. So it was repeated at 15 weeks ga, which showed the alarming aberrations that prompted for confirmation amniocentesis and maternal blood test at 17 weeks. Cytogenetic analysis of the amniocentesis was normal but the maternal blood test confirmed maternal mosaicism tetrasomy 9p. MPLA testing at 21 weeks ga showed heterozygous duplication of 9p subtelomeric region on blood sample, but not the buccal swab. The result indicates somatic mosaicism of 9p duplication in this patient. CffDNA was performed at post delivery day 6 and post delivery day 30 showed similar aberrations as 15 weeks ga, once again confirming true maternal mosaic tetrasomy 9p. If it was placental source, aberration would have resolved after delivery. This is important has we could not test the placenta. This finding was relevant and was elaborated in the revised manuscript. I have clarified the chronological order of these tests, and showed their respective results, in the new version. 

Point 3: The discussion should discuss the results compare to the available literature. 

Available literature is sparse. Some discussion points are derived from our own explanations.

Point 4: 

The conclusion should be expanded and more comprehensive as it was already reflected in the abstract. 

Since the baby was already born, further perinatal information should be included.

Response 4: I will revise accordingly. As we are absolutely certain that the source of the abnormal ccfDNA is maternal, no further genetic test was provided for the baby. The placental biopsy for genetic study would have been helpful to rule out confined placental mosaicism. Unfortunately our midwife soaked it in formalin before biopsy was taken.